# Transcriptomic and cellular decoding of regional brain vulnerability to neurogenetic disorders

Jakob Seidlitz [1,2 ✉], Ajay Nadig[1], Siyuan Liu[1], Richard A. I. Bethlehem[2], Petra E. Vértes[2,3,4], Sarah E. Morgan [2], František Váša[2], Rafael Romero-Garcia [2], François M. Lalonde [1], Liv S. Clasen[1], Jonathan D. Blumenthal[1], Casey Paquola [5], Boris Bernhardt[5], Konrad Wagstyl[2,6], Damon Polioudakis[7], Luis de la Torre-Ubieta[7,8], Daniel H. Geschwind [7,9], Joan C. Han[10,11,12], Nancy R. Lee[13], Declan G. Murphy [14], Edward T. Bullmore[2,15] & Armin Raznahan[1 ✉]

Neurodevelopmental disorders have a heritable component and are associated with region specific alterations in brain anatomy. However, it is unclear how genetic risks for neurodevelopmental disorders are translated into spatially patterned brain vulnerabilities. Here, we integrated cortical neuroimaging data from patients with neurodevelopmental disorders caused by genomic copy number variations (CNVs) and gene expression data from healthy subjects. For each of the six investigated disorders, we show that spatial patterns of cortical anatomy changes in youth are correlated with cortical spatial expression of CNV genes in neurotypical adults. By transforming normative bulk-tissue cortical expression data into cell-type expression maps, we link anatomical change maps in each analysed disorder to specific cell classes as well as the CNV-region genes they express. Our findings reveal organizing principles that regulate the mapping of genetic risks onto regional brain changes in neurogenetic disorders. Our findings will enable screening for candidate molecular mechanisms from readily available neuroimaging data.

[1] Developmental Neurogenomics Unit, National Institute of Mental Health, Bethesda, MD, USA. [2] Department of Psychiatry, University of Cambridge, Cambridge, UK. [3] School of Mathematical Sciences, Queen Mary University of London, London, UK. [4] The Alan Turing Institute, London, UK. [5] McConnell Brain Imaging Centre, Montreal Neurological Institute and Hospital, Montreal, QC, Canada. [6] McGill Centre for Integrative Neuroscience, McGill University, Montreal, QC, Canada. [7] Department of Neurology, Center for Autism Research and Treatment, Semel Institute, David Geffen School of Medicine, UCLA, Los Angeles, CA, USA. [8] Department of Psychiatry and Biobehavioral Sciences, Semel Institute, David Geffen School of Medicine, UCLA, Los Angeles, CA, USA. [9] Department of Human Genetics, David Geffen School of Medicine, UCLA, Los Angeles, CA, USA. [10] Departments of Pediatrics and Physiology, University of Tennessee Health Science Center and Le Bonheur Children's Foundation Research Institute, Memphis, TN, USA. [11] Pediatrics and Developmental Neuropsychiatry Branch, National Institute of Mental Health, NIH, Bethesda, MD, USA. [12] Unit on Metabolism and Neuroendocrinology, Eunice Kennedy Shriver National Institute of Child Health and Human Development, NIH, Bethesda, MD, USA. [13] Department of Psychology, Drexel University, Philadelphia, PA, USA. [14] Institute of Psychiatry, King's College London, London, UK. [15] Cambridgeshire and Peterborough NHS Foundation Trust, Huntingdon, UK. ✉email: jakob.seidlitz@nih.gov; raznahana@mail.nih.gov

Neurodevelopmental disorders such as autism and schizophrenia are highly heritable, and associated with spatially selective changes in brain structure and function[1–3], which remain poorly understood in mechanistic terms. In particular, it remains unclear how genetic risks translate into the spatial patterns of altered brain anatomy that have been reported in neurodevelopmental disorders. Clarifying factors that shape regional brain vulnerability to genetic risks would help to advance the translational medicine of neurodevelopmental disorders. However, progress in this area is complicated by several issues including (i) the etiological heterogeneity of behaviorally defined neurodevelopmental disorders[4], (ii) the vast search space of candidate biological features that could determine regional brain vulnerability[5], and (iii) lack of access to spatiotemporally comprehensive postmortem brain tissue from patients.

Recent experimental work in mice has suggested an organizing principle for regional brain vulnerability to genetic risks that may apply in human neurodevelopmental disorders. Specifically, the spatial patterning of neuroanatomical changes in MRI brain scans from mutant mouse models with disruptions of neurodevelopmental genes can be predicted by intrinsic expression gradients of those genes in the brains of wild-type mice[6,7]. Strikingly, this spatial coupling was recovered using expression data from adult wild-type mice—despite the likely operation of mutant allele effects much earlier in brain development. These murine data therefore propose a transcriptional vulnerability model for the spatial patterning of altered brain anatomy in genetically determined disorders of brain development and further suggest that evidence for this model could be recovered even if intrinsic expression gradients are being measured in adulthood. To date, however, tests of the transcriptional vulnerability model in humans have only been available from studies of brain anatomy patients with idiopathic autism and schizophrenia[8–11]. Because the genetic basis of disease is unknown in idiopathic cases, it is not possible to determine if observed spatial patterns of neuroanatomical change are related to normative expression gradients for the causal genes.

Here, we conduct a genetically informed test of the transcriptional vulnerability model in humans. To achieve this test, we study youth with known genomic dosage variations that increase risk for one or more adverse neurodevelopmental outcomes such as intellectual disability, specific learning disability, autism spectrum disorder, attention deficit hyperactivity disorder and schizophrenia: Down syndrome[12] (trisomy 21), sex chromosome aneuploidies[13] (XO, XXX, XXY, XYY, XXYY), Velocardiofacial syndrome[14] (del22q11) and WAGR syndrome[15] (del11p13). These diverse genetically defined disorders encompass both gains and losses of genetic material, which range in size from subchromosomal copy number variations (CNVs) to full chromosomal aneuploidies (henceforth collectively referred to as CNVs) —thereby providing a powerful test for generalizability of the transcriptional vulnerability model. We also seek to refine and apply this transcriptional vulnerability model, by testing if the intrinsic gene expression gradients hypothesized to guide neuroanatomical disruptions can themselves be rooted in spatial patterning of the human brain by different cell-types. Such grounding of regional transcriptional vulnerability in cell-type composition could provide a principled framework for nominating specific genes within specific cell-types that may account for altered anatomy in a given brain region to a given neurogenetic disorder.

## Results

**Mapping altered cortical anatomy in six different CNV conditions**. We assembled a total of 518 structural magnetic resonance imaging (sMRI) brain scans from matched case −control cohorts spanning eight different neurogenetic disorders: XXX, XXY, XYY, XXYY, trisomy 21 (Down syndrome), X-monosomy (Turner syndrome), del22q11.2 (Velocardiofacial syndrome, VCFS) and del11p13 (Wilms Tumor−Aniridia syndrome, WAGR) (Supplementary Table 1; total $N = 231$ patients, 287 controls). Because the distinct gene sets defining each of these CNV disorders is known, we were able to conduct a series of strict, independent tests of the transcriptional vulnerability model in humans. Specifically, we asked if the map of cortical anatomy change in each of the six CNV states represented by these disorders (henceforth: +X, +Y, +21, −X, −22q11, −11p13) was preferentially correlated with spatial patterns of expression for the known genes that defined that disorder. Each case−control pair was scanned on the same MRI machine using the same image-acquisition parameters.

A map of cortical anatomy change was made for each of the six CNVs vs. matched controls using morphometric similarity (MS) mapping. Rather than considering individual anatomical features such as cortical thickness, area, and curvature, MS mapping combines information across multiple cortical features to estimate a network of MS between pairs of cortical regions within an individual brain. This network can be summarized as a person-specific map of mean MS for each cortical region (relative to all other cortical regions). Group-level comparisons were used to determine the spatial pattern of cortical MS change associated with each CNV (Fig. 1a, Supplementary Fig. 1; see "Methods"). Our use of MS mapping in this study was motivated by two key considerations: (i) CNV disorders have dissociable impacts on different anatomical features of the cortical sheet[1], and MS mapping provides a means of integrating this rich information, (ii) cortical MS gradients are strongly aligned with the molecular and cytological aspects of cortical patterning that we sought to probe in our test of the transcriptional vulnerability model[16]. Further information on topological and network features of MSNs is provided in the original paper[16]. To compare results from the use of MSNs vs. classical single-feature approaches to cortical morphometry, we also generated supplementary maps of anatomical change in each CNV for all of the individual cortical features that are combined in MS mapping (Supplementary Fig. 2a). To test for differential contribution of single cortical features to observed MS changes in each CNV, we recomputed CNV-specific MS change maps with exclusion of each individual cortical feature prior to MSN construction and then determined which of these single-feature exclusions most change the topography of observed MS change for each CNV. This leave-one-out procedure showed that mean curvature (+X, +Y) and gray matter volume (−X, +21,−22q11,−11p13) were the features that most contributed to the topography of observed MS changes (Supplementary Fig. 2b).

Each of the six CNVs studied induced a distinct spatial pattern of MS change across the cortex, with regionally specific MS increases (red) and decreases (blue) relative to healthy control participants (Fig. 1b). The distinctiveness of MS change in each CNV was not an artifact of differences between the cohorts of healthy individuals against which each CNV was being compared (Supplementary Fig. 3a). Observed MS change maps in each CNV were not altered by inclusion of total surface areas as a covariate to capture brain size variation (correlation in MS change across regions with vs. without covariate: +X, $r = 0.99$; +Y, $r = 0.98$; −X, $r = 0.98$; +21, $r = 0.99$; −22q11, $r = 0.97$; −11p13, $r = 0.98$). Supplementary edge-level analyses (i.e. examining CNV effects on inter-regional MS) revealed the distinct patterns of anatomical disruption that underlay regional MS increases vs. MS decreases in CNV carriers (see "Methods", Supplementary Fig. 3b). Comparison of MS change maps with a standard functional-connectivity parcellation of the cortex[17]

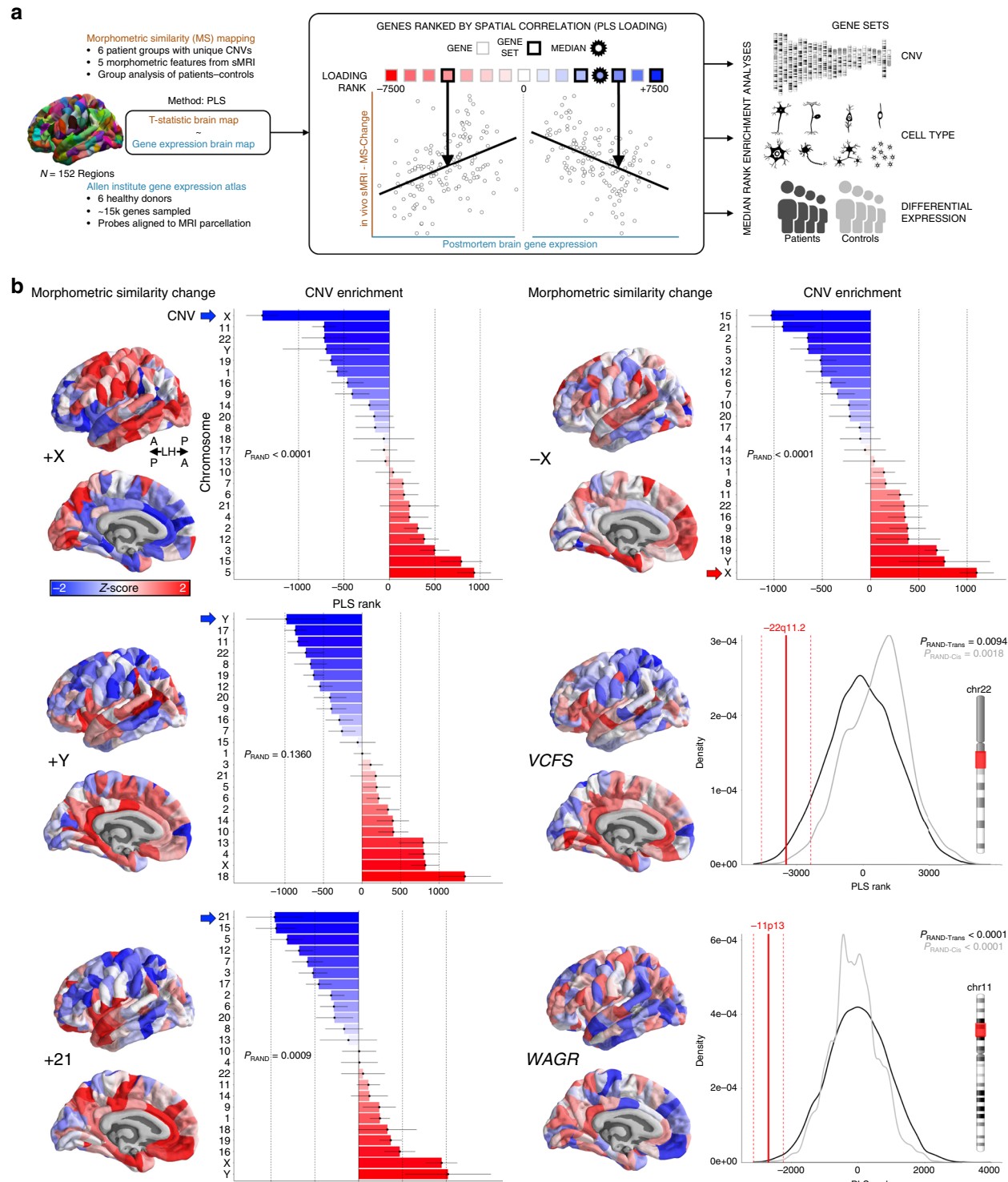

**Fig. 1 Transcriptomic specificity of neuroanatomical effects. a** Schematic outlining the main imaging-transcriptomic enrichment analyses and statistical tests. **b** (left) Surface projections of T-statistics (z-scored for plotting purposes) for CNV effects on regional morphometric similarity (MS). Despite some overlap across CNVs, each CNV induces a distinct profile of MS change. For full chromosome CNVs, neighboring point range plots show the median (point) and standard error (range) rank of each chromosomal gene set—based on gene rankings from the PLS analysis (see (**a**), N genes per chromosome provided in Supplementary Dataset 3). The chromosomal gene set for each CNV possessed a more extreme median rank than all other chromosomal gene sets, and the polarity of this effect was opposite for chromosomal duplications (CNV gene set high ranked) versus deletion (CNV gene set low ranked). For subchromosomal CNVs (depicted as red in the respective chromosome ideograms), density plots show median (solid line) and standard error (dashed line) ranks for the relevant CNV gene set. Observed ranks are shown relative to two null distributions: $P_{RAND-Trans}$ (black), and $P_{RAND-Cis}$ (gray). $P_{RAND}$ was calculated using 10,000 gene rank permutations (black). $P_{RAND-Cis}$ was calculated similarly to $P_{RAND-Trans}$ but only sampling gene ranks from the respective chromosome of the CNV. All permuted P values were not further corrected for multiple comparisons, and were determined based on one-sided tests of gene set enrichment (median rank; see "Methods").

revealed significant variation across CNVs in the magnitude of average MS-change across different functional networks (Supplementary Fig. 2c; ANOVA ($\Delta$MS~CNV + Network + CNV × Network; $F(6)_{Class} = 6.323$, $P = 1.6 \times 10^{-6}$; $F(30)_{Group \times Class} = 3.466$, $P = 2 \times 10^{-9}$). For example, MS increases in participants with Down syndrome relative to controls were significantly enriched within a well-defined ventral attentional network[17].

**Aligning anatomical changes in CNVs with cortical gene expression**. Next, to query the transcriptomic correlates of altered MS in each CNV disorder, we aligned each CNV's MS change map to the same publicly available atlas of gene expression for ~15k genes in adult human cortex from the Allen Human Brain Atlas (AHBA dataset)[18] (see "Methods", Fig. 1b, Supplementary Fig. 1). We used partial least squares (PLS) regression to rank all ~15k genes in this atlas by their multivariate correlation[19,20] with each CNV's MS change map—resulting in one ranked gene list for each CNV disorder (Fig. 1a, Supplementary Dataset 1). In these lists, genes with expression patterns that are more strongly correlated with the corresponding MS change map have large positive or negative PLS weights and therefore occupy more extreme ranks. The polarity of these ranked lists was set so that genes with strongly positive PLS weights occupied extreme low ranks (i.e., closer to c. −7500, Fig. 1a), and showed positive spatial correlations between their cortical expression and cortical MS change in patients vs. controls. Conversely, high-ranking genes (i.e., closer to c. +7500, Fig. 1a) had strongly negative PLS weights, and were expressed in spatial patterns that correlated negatively with MS change in patients vs. controls. These CNV-specific ranked gene lists quantify the degree of spatial correspondence between observed cortical changes in a CNV disorder and cortical expression of that CNVs gene set in health as the median rank position of genes within the CNV region. Null distributions for this median rank test statistic can be generated by gene rank permutation (see "Methods"). Thus, for any given CNV, one can test the transcriptional vulnerability model by asking if the median rank of genes within the causal CNV region is more extreme than would be expected by chance. For subchromosomal CNVs, a null distribution of ranks was created from 10,000 randomly sampled gene sets of the same size from the whole genome ($P_{RAND-Trans}$) as well as from the same chromosome ($P_{RAND-Cis}$). For chromosomal aneuploidies, we harnessed the natural comparison gene sets provided by other chromosomes and primarily asked if the observed median rank of the aneuploidic chromosome was more extreme than that of all other chromosomal gene sets. Gene rank permutations were used to determine the likelihood of seeing the aneuploidic chromosome possess the most extreme median rank of all chromosomes ($P_{RAND}$). To ensure robustness across the six donors in the AHBA, we performed a leave-one-donor-out PLS analysis for each syndrome's brain map, which showed highly consistent PLS loadings across all permutations relative to the PLS loadings derived from the whole AHBA dataset (+X mean $r = 0.98$; +Y mean $r = 0.84$; −X mean $r = 0.97$; +21 mean $r = 0.98$; −22q11 mean $r = 0.96$; −11p13 mean $r = 0.98$; all $N = 15,043$ genes).

Our analyses found independent support for the transcriptional vulnerability model in each of the six CNV conditions studied (Fig. 1b, Supplementary Dataset 2). The omnibus $P$ value for this observation exceeded the limits of our permutation test (i.e. $P < 0.0001$, see "Methods"). In all three CNVs involving abnormal gain of a chromosome (+X, +Y, +21), the relevant chromosomal gene set showed a higher median rank than all other chromosomal gene sets. Conversely, in Turner syndrome, which involves the loss of an X-chromosome (−X), the X-chromosome gene set showed a lower median rank than all other

chromosomal gene sets (correlation between +/− X PLS scores, $r = -0.92$). Furthermore, in the two conditions associated with subchromosomal gene losses (−22q11, −11p13), the CNV gene set also showed a significantly lower median rank than null gene sets of the same size drawn from the whole genome. Thus, for these six different CNV disorders, brain regions showing relatively high expression of the causal gene set in health tended to show MS decreases in patients carrying a duplication of the gene set, and MS increases in patients carrying a deletion of that gene set. Conversely, brain regions showing relatively low expression of the causal gene set in health tended to show MS increases in patients with gene set deletion, and MS increases in gene set duplication. For 5/6 CNVs studied (all but +Y), the above median rank results were statistically significant at Bonferroni-corrected $P < 0.05$. Supplementary analyses clarified that weaker statistical significance of this median rank permutation test for the +Y CNV condition was a predictable consequence of the small number of Y-linked genes with available brain expression data (Supplementary Fig. 2c). MS change maps performed better than individual anatomical feature change maps for recovering the specific relationships between cortical gene expression and anatomical change in each CNV (Supplementary Fig. 2a, Supplementary Dataset 3). Taken together, these findings provide strong evidence that the transcriptional vulnerability model is a relevant general organizing principle for spatial patterning of anatomical changes in genetically defined neurodevelopmental disorders.

Our integration of neuroimaging and transcriptomic data also provided several biological insights into each of the individual CNVs studied. First, we were able to define those genes within each CNV region that were expressed in spatial patterns which most closely resembled CNV-induced anatomical changes (Supplementary Dataset 1). For example, the most-extreme-ranking CNV region gene relative to anatomical change in each CNV was; +X: *ZCCHC12*, +Y: *EIF1AY*, −X: *GABRA3*, +21: *PCP4*, −22q11: *MAPK1*, −11p13: *TRIM44*. This ranking by spatial correspondence provides an especially useful criterion for prioritization of genes within large chromosomal and subchromosomal CNVs. Second, ranked gene lists also identified genes outside the CNV region with expression patterns that most closely mirrored observed anatomical changes—suggesting candidate molecular partners that might interact with altered expression of CNV genes to shape regional brain vulnerability (Supplementary Dataset 1). Third, rank-based GO term enrichment analyses identified biological process and cellular component annotations that were overrepresented at extremes of each CNV disorder's gene list (Supplementary Dataset 4). For example, for +21, genes with spatial expression related to MS increases showed enrichment for cell communication and synapse composition, whereas genes with spatial expression related to MS decreases showed enrichment for ion transport and intracellular structures (Supplementary Dataset 4).

**Deriving cell-class gene expression gradients in the human cortex**. The above findings indicate that intrinsic transcriptomic differences across the cortical sheet in adulthood are correlated with regional anatomical vulnerability of the cortical sheet to genetically defined neurodevelopmental disorders. However, regional differences in cortical gene expression across the lifespan are themselves thought to largely reflect regional differences in cellular composition of the cortical sheet[21]. We therefore reasoned that the spatial correspondence between expression of CNV genes in health and anatomical changes in CNV carriers (Fig. 1) may be underpinned by patterned expression of CNV genes across different cell-types with varying spatial distributions.

As there are no spatially comprehensive maps of cell-type density across the human brain with which one could test this hypothesis, we generated cell-class density proxy maps from bulk-tissue AHBA expression data using information from single-cell gene expression studies. To generate this reference set of cell-type maps in the human cortex, we compiled cell-specific gene sets ($N = 58$) from all available large-scale single-cell studies of the adult human cortex (see "Methods", Supplementary Dataset 5), and then calculated the mean expression of each cell-type gene set in each of the 152 cortical regions within our MRI-registered projection of the AHBA dataset (see "Methods"). Unsupervised hierarchical clustering of this cell-by-region expression matrix using the gap-statistic criterion[22] distinguished three broad cell groups with distinct patterns of regional expression (Fig. 2b): (i) oligodendrocytes, (ii) other glial and endothelial cells, and (ii) excitatory and inhibitory neurons. Further co-clustering of cells within these three groups by the similarity in their regional expression profiles (Fig. 2b; see "Methods") recovered all seven canonical cell classes within the central nervous system: microglia, endothelial cells, oligodendrocyte precursors (OPCs), oligodendrocytes, astrocytes, excitatory and inhibitory neurons. Thus, independently derived cell-type gene sets from single-cell genomics, reflecting diverse cortical tissue samples and varying analytic methods from five separate studies, are perfectly grouped by cell class using the sole criterion of similarity in their regional expression across bulk samples of cortical tissue.

We generated a single omnibus gene set for each of these seven cell classes by collapsing across study-specific gene sets, and we then visualized the mean expression for each cell-class gene set across the cortex (Fig. 2b; see "Methods"). These transcriptomic proxy maps for cellular patterning across the human cortex could be validated against several independently generated maps of cortical microstructure from neuroimaging and histology (see "Methods", Supplementary Fig. 4a). For example, (i) the oligodendrocyte cell-class expression map showed a statistically significant positive correlation with a map of intracortical myelination as indexed by in vivo magnetization transfer imaging[19], whereas (ii) the astrocyte cell-class expression map showed statistically significantly positive correlations with several histological and neuroimaging markers for associative cortices with expanded supragranular layer thickness[23–25]. Furthermore, for two of our cell-class expression maps, we harnessed available cell-class-specific markers to test for convergent validity with available in situ hybridization data (ISH). Specifically, we examined *GFAP* and *MBP* staining intensity as ISH markers for astrocytes and oligodendrocytes (respectively) in postmortem slices from cortical regions showing opposite patterns of astrocyte and oligodendrocyte gene expression in our transcriptomic cell-class proxy maps (Fig. 2c). This analysis revealed a close congruence between regional cell-class representation from our deconvolution approach (Fig. 2c), and cell-class representation from direct ISH staining (Supplementary Fig. 4b).

**Aligning anatomical changes in CNVs to cell-class gene expression**. We used our derived cell-class gene sets to test if observed cortical anatomy changes in each CNV disorder were organized with respect to broad cell-class gradients in the human cortical sheet. We achieved this test for each CNV disorder by screening for cell classes, which (i) had a cell-class gene set that possessed a significantly extreme median rank ($P_{RAND} < 0.05$) in the CNV's ranked gene list from AHBA alignment (Supplementary Dataset 1), and (ii) included one or more extreme-ranking (i.e., top/bottom 5% PLS ranks) genes from the CNV region within their cell-class gene set (Supplementary Dataset 5). These criteria identified several pairwise associations between

spatial patterns of cortical anatomy change in CNVs and expression gradients of cell classes expressing genes from within the CNV region. Some of these cell-gene associations integrated cellular and molecular findings from prior research—providing (i) evidence of convergent validity between the results of our analytic approach and prior work, and (ii) a parsimonious integration of previously disconnected findings (Supplementary Fig. 4c). For example, oligodendrocyte precursor cells and the chromosome 21 gene *NCAM2* have both been separately implicated in the neurobiology of Down syndrome[26–28]. Our analytic method recovered and synthesized these prior associations by showing that cortical MS increases in Down syndrome are preferentially localized to cortical regions with a strong expression signature for OPCs, which include *NCAM2* in their cell-class gene set (Supplementary Fig. 4c). We also identified informative cell-gene associations for regions of MS change in all other CNVs examined (excepting +Y), including: MS decreases in Down syndrome and the transcriptomic signature for *PCP4*-expressing oligodendrocytes; MS increases in WAGR syndrome ($-11p13$) and *PAX6*-expressing astrocytes; MS increases in VCFS syndrome ($-22q11$) and *MAPK1*-expressing inhibitory neurons; MS changes in X-chromosome aneuploidies and expression of oligodendrocytes, endothelial cells and astrocytes (which possess cell-class gene sets that include neurodevelopmentally pertinent X-linked genes such as *AMMECR1, ITM2A* and *PTCHD1*). Importantly, our analytic framework can narrow hypotheses regarding cell-specific drivers of regions of altered brain development in each CNV without requiring access to postmortem brain tissue from patients.

**Validation against gene expression data from CNV carriers**. Collectively, the above findings provide strong evidence that the spatial patterning of altered brain anatomy in pathogenic CNV disorders in humans is organized by intrinsic expression gradients of CNV-region genes in the human brain. We next sought to further validate this transcriptional vulnerability model by testing two associated predictions against direct measures of altered gene expression in patients. First, we predicted that the ranked gene list for each CNV disorder from comparison of neuroimaging and AHBA data (Supplementary Dataset 1) should differentiate between (i.e. differently rank at $P_{RAND} < 0.05$) those CNV region genes that do show robust expression changes in patients [i.e. dosage-sensitive (DS) genes], and those that do not (nDS genes). This prediction held for all three CNV disorders with available DS and nDS gene sets from genome-wide comparisons of expression between patient and control tissues: +21, +X and -X (see "Methods", Fig. 3b, Supplementary Dataset 6)[29–31]. Strikingly, we observed statistically significant differential ranking of DS and nDS gene sets ($P_{RAND} < 0.05$) regardless of whether the gene sets had been defined in postmortem brain tissue (available for +21), or blood-derived lymphoblastoid cell lines (LCLs, available for +21, +X and −X). The fact that CNV region gene dosage sensitivity in patient LCLs is predictive of imaging−transcriptomic associations in brain tissue is consistent with the idea that *cis*-effects of a CNV on gene expression may be highly reproducible across tissue types (even though normative gene expression profiles vary greatly between different tissue types[32]). This idea of cross-tissue stability in *cis*-effects of a CNV on gene expression is supported by prior research in model systems[33], and our own observation of a statistically significant correlation across genes between the magnitude of expression change for chromosome 21 genes in brain tissue vs. in LCLs from patients with Down syndrome ($r = 0.34$, $P_{RAND} = 0.04$, see "Methods").

For all three CNVs considered, median rank differences between DS and nDS gene sets were driven by a small subset of

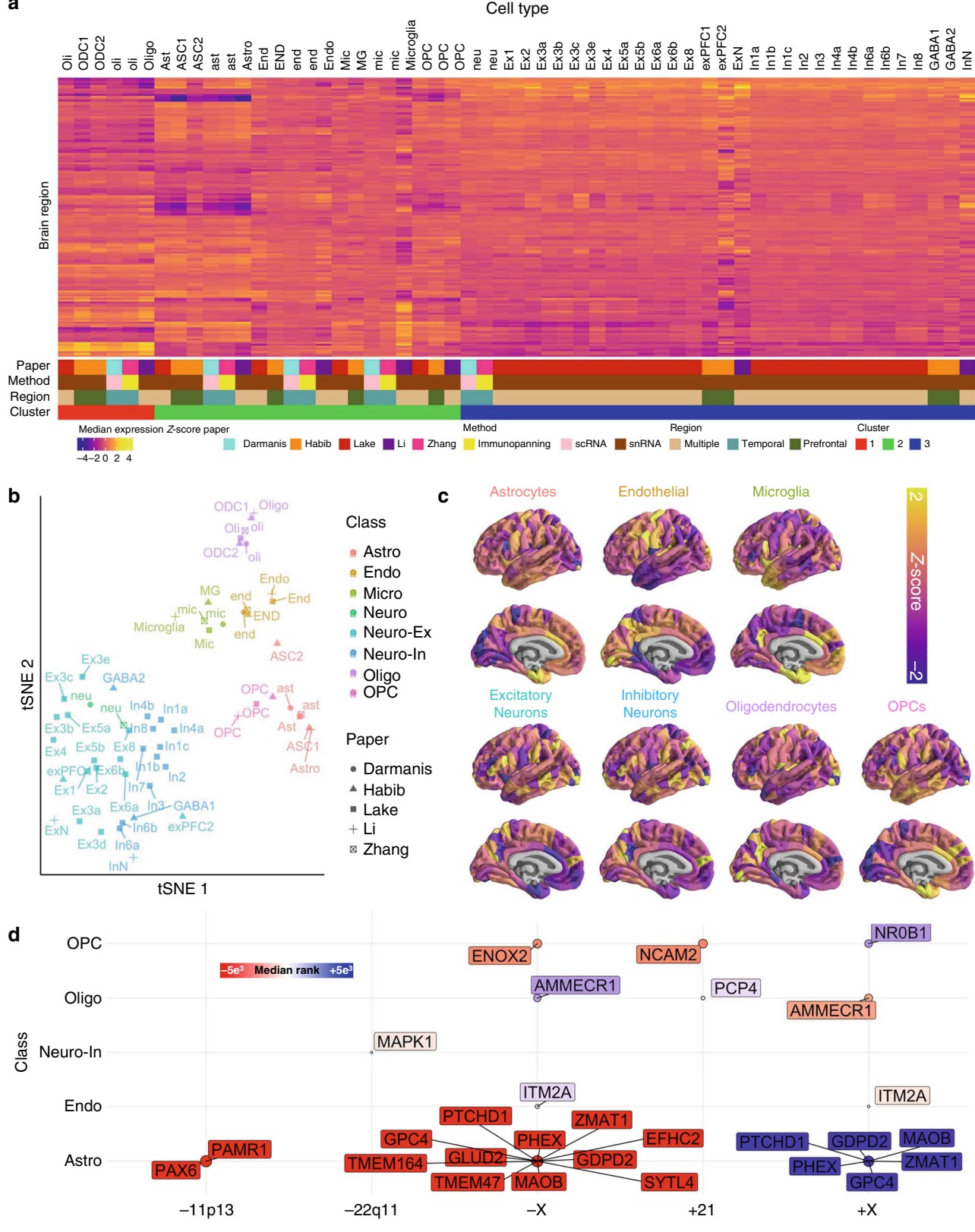

DS genes (DS$^{SS}$, Fig. 3b, c, rank decile analysis; see "Methods"). These DS$^{SS}$ genes possessed strongly positive PLS weights relative to cortical changes in +21 and +X = indicating that they are most highly expressed in cortical regions of MS increase in patients vs. controls (e.g., +21: insula and cingulate cortex, +X: anterior cingulate). Conversely, for cortical changes in −X, DS$^{SS}$ genes possessed strongly negative PLS weights— indicating that associations between regional DS$^{SS}$/nDS gene

precuneus, lateral temporal lobe), and that they are least expressed in regions of MS reduction (e.g., +21: fronto-parietal areas, +X: anterior cingulate). Conversely, for cortical changes in −X, DS$^{SS}$ genes possessed strongly negative PLS weights— indicating that associations between regional DS$^{SS}$/nDS gene

**Fig. 2 Cell type decoding of AHBA microarray and CNV gene ranks. a** Regional median expression (*Z* score) in the AIBS microarray dataset of cell-specific gene sets, aggregated across five single-cell sequencing studies and ordered according to hierarchical clustering (*N* = 3 clusters based on the gap statistic). Cell type abbreviations are maintained from the original study (see also Supplementary Dataset 5). **b** T-distributed stochastic neighborhood embedding (tSNE) of cell-specific gene sets based on their spatial expression profiles distinguishes seven canonical cell classes (color coded). **c** Regional weighted expression maps (see "Methods") of each canonical cell class from (**b**). **d** Significant associations between cell classes and MS change in different CNVs. Circles indicate cell classes with gene sets that show statistically median rank enrichment (one-sided test) relative to PLS-derived ranked gene lists for each CNV disorder (*P*$_{RAND}$ < 0.05). Circles color indicates the direction of median rank enrichment: red circled cell classes show high expression in brain regions where MS is greater in patients than controls (vice versa for blue circles). Named genes for each cell class are (i) expressed by the cell, (ii) in the respective CNV, and (iii) highly correlated with regional variation in MS change for that CNV (i.e. in the top 5% of PLS ranks).

expression and MS changes in Turner syndrome are a mirror image of those in +X states. Thus, for all three CNV conditions considered, the spatial patterning of cortical MS changes was preferentially correlated with the patterned expression of CNV genes, but in opposite directions for DS$^{SS}$ vs. nDS gene sets (i.e., Fig. 1b vs. Fig. 3b, respectively). These observations implied that the relative expression of DS$^{SS}$ vs. nDS genes should provide a strong predictor of regional MS change in these neurodevelopmental disorders. This inference was verified for all three CNVs using a surface-based rotational test to compare the map of observed anatomical changes in the CNV disorder to that predicted by an index of DS$^{SS}$ vs., nDS expression from the AHBA (*P*$_{SPIN}$ < 0.001, Fig. 3d).

As a second validation from patient expression data, we reasoned that if (1) the spatial patterning of MS changes is organized by intrinsic gradients of gene expression in the human cortex (Figs. 1b and 3d), and (2) there is a causal relationship between cortical MS changes and altered expression of CNV-region genes in CNV carriers, then (3) CNV carriers with greater dysregulation of DS genes should show a more pronounced MS changes along the spatial gradient that coheres with intrinsic cortical expression of CNV genes. As there are no available cohorts of CNV carriers that have provided both in vivo neuroimaging and postmortem gene expression data from brain tissue—we sought to test this prediction by leveraging the fact that proximal effects of a CNV on expression of CNV region genes are known to show good stability across tissues (see above).

A subset of 55 patients in our study that carried an extra X-chromosomes had previously provided a blood sample for gene expression analysis (*N* = 55, karyotypes: XXX, XXY, XXYY). These blood samples had been used to make LCLs, from which we had measured expression for 11 DS X-chromosome genes by qPCR[29] (see "Methods"). To interrelate peripheral gene expression and cortical MS across individuals, we (i) scaled regional cortical MS and gene expression values within each karyotype group (to remove potential between-karyotype effects), (ii) used PLS regression to define the principal component of shared variance between LCL gene expression and cortical MS (Fig. 4a). This procedure identified a statistically significant component of shared variance between MS change and DS gene expression across patients (*P* = 0.0094, see "Methods"). Strikingly, the cortical region loadings for this component closely recapitulated the spatial gradient of MS change associated with carriage of an extra X chromosome (*r* = 0.59, *P* < 0.0001 by test of random spatial rotation of cortical maps and by resampling patients; see "Methods", Fig. 4b). Thus, CNV-induced changes in cortical anatomy are not only coupled to regional variation in the cortical expression of CNV genes in health (Fig. 1), but also to interindividual variation in the degree of altered expression in CNV region genes among CNV carriers (Fig. 4).

## Discussion

The methods and results presented above offer several theoretical and empirical inroads into the biology of neurogenetic disorders

First, by studying genetically defined (rather than behaviorally defined) patient cohorts, we test the transcriptional vulnerability model in humans by benchmarking findings against a ground-truth set of genes that are known in advance and define the disorders studied. In this way, our analyses in humans are analogous to gold-standard tests of the transcriptional vulnerability model that have so far only been possible in transgenic mice with experimentally induced CNVs[34]. We find that the transcriptional vulnerability model is indeed an organizing principle for the spatial targeting of pathogenic CNV effects on human brain anatomy (Fig. 1). Strikingly, in humans, as in mice[34], these spatial relationships between intrinsic cortical expression of CNV genes and cortical anatomy changes in CNV carriers could be recovered despite reliance on spatially comprehensive gene expression data that had been derived from adult brains. Thus, although gene expression landscapes within the brain show profound spatio-temporal dynamism[21], there appears to be sufficient stability in the topology of gene expression to recover CNV-specific associations over developmental time.

Second, we show that cellular organization of the human brain provides a biological lens that can translate disease-related alterations of specific genes into disease-related alterations of specific distributed brain regions. We exploit this cellular framework to narrow hypotheses about the specific genes and cell-types that might underpin regional cortical disruptions in patients with Down, VCFS, WAGR and sex chromosome aneuploidy syndromes (Fig. 2). Critically, the analytic workflow we pursue achieves a broad screen of many potential brain regions, cell classes and genes without relying on postmortem tissue from patients or generalization from model systems. This approach provides a practical advantage given the scarcity of postmortem brain tissue from patients (especially those with rare genetic disorders), and also enables us to make predictions regarding the biology of distinctly human disorders using data from native human tissue. To facilitate wider use of this cell-map decoding approach, we are publishing (i) the cell-class gene sets derived by our spatial integration of prior single-cell gene expression data (Supplementary Dataset 2), and (ii) the spatially comprehensive cortical maps for expression of each cell-class gene set in standard neuroimaging space (see "Data availability").

Third, we refine and further validate the transcriptional vulnerability model using direct measures of gene expression in CNV carriers. Specifically—considering full CNVs of chromosomes X and 21—we establish that relationships between cortical expression in health and cortical anatomy change in CNV carriers are significantly different for CNV region genes with contrasting dosage sensitivity in patient tissue. As a consequence, both the valance and magnitude of regional cortical anatomy change in CNV carriers could be predicted from a single index of regional cortical gene expression in health—which contrasted relative expression of those CNV genes that do show dosage sensitivity in CNV carriers vs. those that do not (Fig. 3). Notably, this finding held whether the expression data used to define CNV gene dosage sensitivity status were derived from brain tissue or from LCL

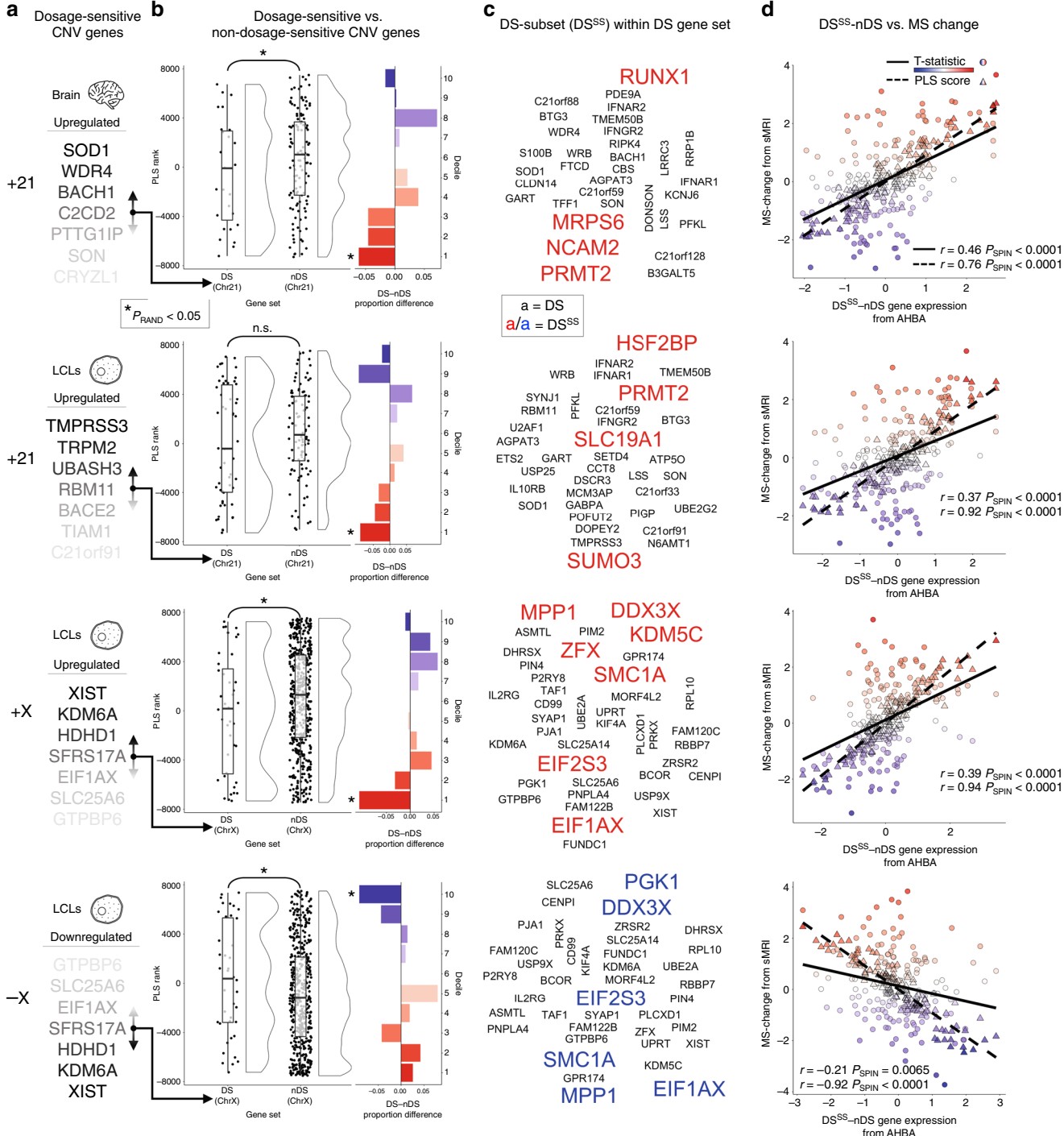

**Fig. 3 CNV gene dosage sensitivity predicts spatial coupling of gene expression brain anatomy. a** Top dosage-sensitive (DS) genes in brain tissue and blood-derived lymphoblastoid cell lines (LCLs) from CNV carriers (brain: +21. LCLs: +21,+X, −X, see "Methods"). **b** Raincloud plots showing the different distributions of ranks for DS and non-DS (nDS) genes (N genes per DS/nDS set provided in Supplementary Dataset 5). Boxplots show the median, interquartile range (IQR), and whiskers (1.5× IQR). Neighboring barplots show decile-specific differences in proportions of DS vs. nDS genes. The statistically significant (*, $P_{RAND} < 0.05$) median rank differences between DS and nDS gene sets are driven by a subset of DS genes (DS$^{SS}$), which are significantly enriched at extreme ranks. **c** DS$^{SS}$ gene names highlighted from the DS gene set. **d** Spatial correlations between DS$^{SS}$−nDS differential gene expression and both regional PLS scores and regional MS change for each CNV. Colors represent high (blue) and low (red) rank. All permuted P values were not further corrected for multiple comparisons, and were determined based on one-sided tests of gene set enrichment (median rank; see "Methods").

tissue. Thus, we show that there is not only a close correspondence between the spatial distribution of altered cortical anatomy in CNV carriers and intrinsic expression of CNV-region genes in the human brain (Fig. 1), but that this correspondence can itself

be further refined by dividing CNV region genes by the extent to which their expression is altered in CNV carriers (Fig. 3).

Finally, using paired measures of brain anatomy and gene expression in a large cohort of patients carrying an extra

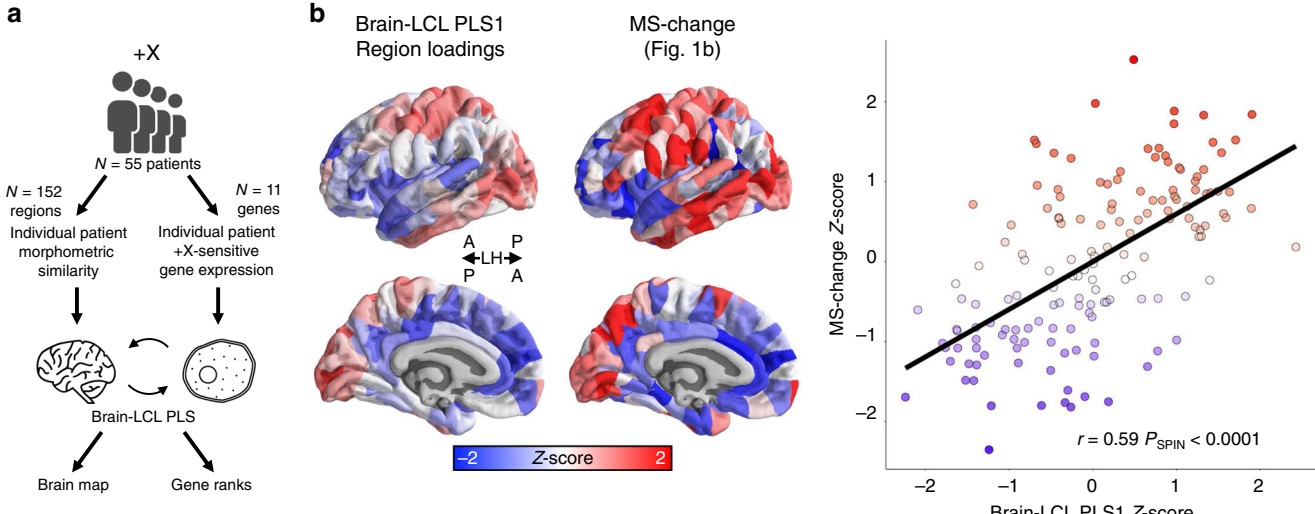

**Fig. 4 Altered gene expression predicts regional anatomical change across patients. a** Schematic outlining the analytic approach to interrelating cortical MS changes from MRI and DS gene expression from LCLs in +X patients. **b** (left) Regional loadings for principal component of shared variance between MS change in brain and DS gene expression in LCLs. (middle) Regional MS change in +X patients compared to controls (from Fig. 1a). Spatial similarity between these maps indicates that +X patients with greater dysregulation of DS genes in blood show a more pronounced manifestation of the +X MS change map. (right) This spatial similarity is quantitatively strong ($r = 0.59$) and statistically significant ($P_{SPIN} < 0.0001$).

X-chromosome, we establish that CNV carriers with greater upregulation of dosage-sensitive X-chromosome genes in blood-derived LCLs show more severe profiles of cortical anatomy change in vivo (Fig. 4). This finding uses the axis of inter-individual variation to provide orthogonal validation of the transcriptional vulnerability model, but also establishes a potentially useful predictive relationship between peripheral gene expression changes and central neuroanatomical changes in CNV carriers. Importantly, the existence of this predictive relationship is fully compatible with well-established differences in gene-expression between blood and brain tissues in health[32]. Rather, our observations only require cross-tissue stability in the proximal impact of a CNV on expression of CNV region genes, and this phenomenon already has good empirical support from studies in different tissues of aneuploidic plants[33], as well as from our own comparison of altered expression of chromosome 21 genes in brain and LCL tissues from patients with Down syndrome (see above). Future work examining the coordinated impact of other CNVs on brain anatomy and gene expression, as well as in other idiopathic neurodevelopmental disorders[35,36], will be critical in substantiating the generalizability of these findings. In general, however, prediction of neuroanatomical phenotypes in CNV carriers from easily gathered measures of peripheral gene expression represents an important proof-of-principle that could potentially open up new avenues towards advances in personalized medicine for CNV-based brain disorders.

In summary, our study adopts a genetic-first approach to provide quasi-experimental support in humans that the spatial patterning of altered brain anatomy in neurogenetic disorders is organized by normative expression gradients of disease-relevant genes in the human brain. We further show that this transcriptional vulnerability model for prediction of regional vulnerability can be linked to cell-type-dependent patterning of gene expression, and validated against direct measures of gene expression in patients. The methods and results we present provide biological insights into several of the specific neurogenetic disorders studied, as well as a framework for transcriptomic and cellular decoding of brain disorders from in vivo neuroimaging data. Crucially, despite not requiring access to any postmortem brain tissue from

patients, or inference from model systems, the methods we present can screen the large multidimensional search space of brain regions, cell types, and genes to propose highly specific mechanistic targets for neurogenetic disorders of the developing human brain.

## Methods

**Cohorts, diagnostic classification, and MRI acquisition.** Sex chromosome aneuploidies [National Institutes of Health—Bethesda, USA (NIH)]: This dataset has been described in detail previously[37–39]. Briefly, we included 297 patients with various supernumerary X- and/or Y-chromosome counts and 165 healthy controls (79 females) (Supplementary Table 1). Patients were recruited through the National Institutes of Health (NIH) website and parent support groups. The presence of sex chromosome aneuploidy was confirmed by karyotype testing. Exclusion criteria included a history of head injury, neurological condition resulting in gross brain abnormalities, and mosaicism (determined by visualization of 50 metaphase spreads in peripheral blood). Healthy controls were enrolled in longitudinal studies of typical brain development[40]. Exclusion criteria for controls included the use of psychiatric medication, enrollment in special education services, history of mental health treatment, or prior diagnosis of a medical condition that impacts the nervous system. Full-scale IQ was measured with the WASI. Subjects were scanned on a 1.5 T GE Signa scanner (axial slices = 124 × 1.5 mm, TE = 5 ms, TR = 24 ms, flip angle = 45°, acquisition matrix = 256 × 192, FOV = 24 cm) using a spoiled-gradient recalled echo (3D-SPGR) imaging sequence. The research protocol was approved by the institutional review board (IRB) at the National Institute of Mental Health, and informed consent or assent was obtained from all individuals who participated in the study, as well as consent from their parents if the child was under the legal age of majority (Clinical trial reg. no. NCT00001246; clinicaltrials.gov).

Down Syndrome/Trisomy 21 [NIH]: This dataset has been described in detail previously[2]. Briefly, we included 26 patients (13 females) with Down Syndrome and 42 healthy controls (21 females) (Supplementary Table 1). All participants with DS had a chromosomal diagnosis of Trisomy 21 according to parent report or direct testing, with no instances of mosaicism. In addition to the genetic inclusion criteria, participants were also required to be free of any history of acquired head injury or other conditions that would cause gross brain abnormalities. Full-scale IQ was measured as follows: for participants under the age of 18, the Differential Ability Scales, Second Edition[41] was administered, and for participants 18 and older, the Kaufman Brief Intelligence Test, Second Edition[42] was administered. Imaging was completed without sedation on the same 3-T General Electric Scanner using an 8-channel head coil. High-resolution (0.94 × 0.94 × 1.2 mm) T1-weighted images were acquired utilizing an ASSET-calibrated magnetization-prepared rapid gradient echo sequence (128 slices; 224 × 224 acquisition matrix; flip angle = 12°; field of view [FOV] = 240 mm). The research protocol was approved by the IRB at the National Institute of Mental Health, and informed consent or assent was obtained from all individuals who participated in the study, as well as consent from

their parents if the child was under the legal age of majority (Clinical trial reg. no. NCT00001246; clinicaltrials.gov).

Wilms Tumor−Aniridia Syndrome (WAGR) [NIH]: A total of 31 patients with heterozygous contiguous gene deletions of incremental variable length on the short arm of chromosome 11 (11p13 deletion), and 23 healthy controls participated in a comprehensive genotype/phenotype study approved by the NIH IRB and with the informed consent of their parents/legal guardians (Supplementary Table 1). Healthy controls were screened and excluded for history of neurological and psychological impairments. Chromosome deletions were characterized by microsatellite marker analysis and oligonucleotide array comparative genomic hybridization. Neuropsychological assessments were conducted using standardized psychological tests. All participants underwent MRI structural brain imaging. Imaging quality controls included visual inspection of the raw images for motion artifacts as well as the quality of the surface and volume segmentations. The image processing results were inspected for surface and volume segmentation errors by FML and AR. The MRI brain scans were collected at 1-mm³ resolution using a 3D TFE T1-weighted sequence on a 3.0 T Philips Achieva MRI scanner equipped with an 8-channel phased array head coil. The sequence parameters were as follows: TR = 8.3 ms, TE = 3.8 ms, TI delay = 1031 ms, 160 shots. In total, 171 slices were acquired in the sagittal plane with an acquisition matrix of 240 × 240 and an FOV of 240 mm. The research protocol was approved by the IRB at the National Institute of Mental Health, and informed consent or assent was obtained from all individuals who participated in the study, as well as consent from their parents if the child was under the legal age of majority (Clinical trial reg. no. NCT00758108; clinicaltrials.gov).

Turner Syndrome (X-monosomy) [Institute of Psychiatry, Psychology and Neuroscience—London, UK (IoPPN)]: This cohort and associated data have been described in depth previously[43,44]. We included 20 females with X-monosomy (Turner's Syndrome) (TS) and 36 healthy controls in this study (Supplementary Table 1). Briefly, participants with TS were recruited through a university-based behavioral genetics research program run in collaboration with the South London and Maudsley NHS Foundation Trust and typically developing controls through local advertisement. Karyotype was determined for each participant with TS by analyzing 30 metaphase spreads using conventional cytogenetic techniques. No participants suffered from any psychiatric or medical disorders that would grossly affect brain function (e.g. epilepsy, neurosurgery, head injury, hypertension, schizophrenia) as determined by a structured clinical interview and examination, as well as review of medical notes. Structural MRI data were acquired using a GE Signa 1.5 T Neuro-optimized MR system (General Electric, Milwaukee, Wisconsin). Whole-head coronal 3D-SPGR images (TR = 14 ms, TE = 3 ms, 256 × 192 acquisition matrix, 124 × 1.5 mm slices) were obtained from all subjects. Ethical approval was obtained from the local Ethics Committee and informed written consent was obtained from all participants (specific Ethics Committee reference ID was held by the since restructured Institute of Psychiatry Ethics Committee. Presence of active Ethics Committee approval verified through original publications[43,44]).

Velocardiofacial Syndrome (VCFS) [IOPPN]: Briefly, all patients with VCFS and control subjects were screened for medical conditions affecting brain function by means of a semi-structured clinical interview and routine blood tests[45,46]. Full-scale intelligence was measured by means of the Canavan et al. shortened version of the Wechsler Adult Intelligence Scale—Revised[47]. We included 27 controls (11 females) alongside 29 participants (13 females) with clinical features of VCFS (Supplementary Table 1) and a 22q11.2 deletion detected by fluorescence in situ hybridization (FISH; Oncor Inc, Gaithersburg, MD, USA). Subjects were scanned on a 1.5 T GE Signa scanner at the Maudsley Hospital in London, UK. A whole-head 3D-SPGR image was acquired for each subject (TR = 11.9 ms; TE = 5.2 ms; 256 × 192 acquisition matrix; 124 × 1.5 mm slices). Ethical approval was obtained from the local Ethics Committee. All subjects (or their guardians, when subjects <16 years old) gave written informed consent after the procedure was fully explained (specific Ethics Committee reference ID was held by the since restructured Institute of Psychiatry Ethics Committee. Presence of active Ethics Committee approval verified through original publications[45,46]).

**Image quality control**. Each of the patient/control datasets used in the current manuscript were taken from previous studies. As such, previous quality control procedures for each dataset can be found in the original papers (see sections above). In addition, each cortical surface reconstruction was manually inspected for topological defects, scrambling patients, and controls to avoid bias.

**Generation of morphometric similarity networks**. All T1-weighted (T1w) scans were processed using the Montreal Neurological Institute's CIVET pipeline[48] (v1.1.10). Due to the lack of multimodal imaging, only (gray matter) morphometric features derived from the T1-weighted scans were estimated (CT cortical thickness, SA surface area, GM gray matter volume, MC mean curvature, IC intrinsic curvature). GM values were estimated using the T1w volumes of each subject. Vertex-wise CT and SA values were estimated using the resulting pial surface reconstructions from CIVET, while MC and IC metrics of these surfaces were estimated using the freely available Caret5 software package[49]. These surface meshes (~80,000 vertices per mesh) were down sampled into our regional parcellation (below), where the vertex-wise estimates of the features were averaged within a

given region in the parcellation. Cortical surface representations were plotted using BrainsForPublication v0.2.1 (https://doi.org/10.5281/zenodo.1069156).

For each subject, regional morphometric features (CT, SA, GM, MC, and IC) were first scaled (Z-scored, per feature across regions) to account for variation in value distributions between the features. After normalization, morphometric similarity networks (MSNs) were generated by computing the regional pairwise Pearson correlations in morphometric feature sets, yielding an association matrix representing the strength of MS between each pair of cortical areas[16]. For all individuals, regional MS (i.e., nodal similarity) estimates were calculated as the average MS between a given cortical region and all others. We have previously demonstrated that there is an extremely high spatial concordance ($r = 0.91$) between the topography of regional MS derived from T1-weighted MRI data alone, and regional MS from more modalities (e.g. a combination of T1w and diffusion weighted imaging[16]).

**Cortical parcellation**. We generated a 308-region ($N = 152$ LH regions) cortical parcellation using a back-tracking algorithm to restrict the parcel size to be approximately 500 mm², with the Desikan−Killiany atlas boundaries as starting points[50,51]. This parcellation has been used in previous structural[19,52,53] and functional[20] imaging studies of connectomes, and was also used in our first study of MSNs[16].

**Statistical analyses of MSN differences**. For each cohort, group-wise effects of disease on nodal similarity were modeled using the "lm" base function in R, with sex and age included as covariates. Linear regression was conducted using the standard ordinary least squares (OLS) procedure. This model was fitted for each region, and the two-sided $T$-statistic (contrast = patient − control) was extracted (represented in Fig. 2a as a $Z$-score for plotting purposes). For the SCA groups, we collectively modeled each chromosome dosage effect as follows:

$$\text{NS}_i \sim \text{intercept} + \beta1(\text{age}) + \beta2(\text{sex}) + \beta3(\text{Xan}) + \beta4(\text{Yan}), \quad (1)$$

where $\text{NS}_i$ is the nodal similarity estimate across subjects at region $i$, and Xan and Yan are the number of supernumerary X and Y chromosomes (respectively). This was done after ruling out any significant interactions between Xan and sex, or Xan and Yan for variation in nodal similarity[1].

For the +21, −X, −22q11.2, −11p13 patient−control comparisons in nodal similarity ($\text{NS}_i$), the following model was used:

$$\text{NS}_i \sim \text{intercept} + \beta1(\text{age}) + \beta2(\text{sex}) + \beta3(\text{Dx}), \quad (2)$$

where Dx is the binary classification of patients and controls.

These procedures resulted in MS change maps for six different CNV conditions, which were taken into subsequent analyses (+X, +Y, +21, −X, −22q11, −11p13).

**Interpreting regional morphometric similarity differences**. Due to the zero-centered nature of the regional MS distribution (Supplementary Fig. 3a), we annotated the regional MS change maps ($T$-statistics) to determine the underlying effects at the edge level (i.e., at the level of inter-regional MS). For each CNV, we first computed the edgewise MS change between patients and controls (i.e., Eqs. (1) or (2) for each edge, or pairwise correlation). Then, for the top ten positive (red in Fig. 1b) and ten negative (blue in Fig. 1b) regional MS $T$-statistics, we took the absolute sum of their corresponding edge $T$ values for each of four possible types of edge effect:

hypercoupling = an edge with a positive weight in controls, and a positive edge $T$-statistic for the CNV effect (i.e. regions that are morphometrically similar in controls being rendered more similar by the CNV)

dedifferentiation = an edge with a negative weight in controls, and a positive edge $T$-statistic for the CNV effect (i.e. regions that are morphometrically dissimilar in controls being rendered less dissimilar by the CNV)

decoupling = an edge with a positive weight in controls, and a negative edge $T$-statistic for the CNV effect (i.e. regions that are morphometrically similar in controls being rendered less similar by the CNV).

hyperdifferentiation = an edge with a negative weight in controls, and a negative edge $T$-statistic for the CNV effect (i.e. regions that are morphometrically dissimilar in controls being rendered more dissimilar by the CNV)

These four effects are depicted in the legend of Supplementary Fig. 3b.

**Derivation of gene sets for each CNV**. Assignments of AHBA genes to chromosome locations were made according to those from ref.[54]. These assignments defined the gene sets used for all chromosome-level analyses. Gene sets for the two subchromosomal CNVs in our study were defined as follows. The 11p13-deletion (WAGR) gene set was defined using the known distribution of proximal and distal breakpoints in the WAGR patient cohort studied (relative to the NCB136/hg18 genome assembly, referenced via the USCS Genome Browser). We used the median proximal and distal breakpoints across patients to define a representative

chromosomal segment for use in analysis, which encompassed 45 AHBA genes in total (Supplementary Dataset 1) including both WAGR critical region genes (*WT1* and *PAX6*). As patient-specific breakpoint data were not available for the 22q11.2-deletion (VCFS) cohort, we defined the gene set for this CNV using reference breakpoints for the most common A−D deletion type (seen in >85% of patients)[55], which encompassed 20 genes from the AHBA dataset.

**Transcriptomic alignment of neuroimaging data.** Methods for the alignment of the microarray gene expression data from six adult human donors, provided by the Allen Human Brain Atlas (AHBA), to the left hemisphere ($N = 152$ regions) of our parcellation has been described in depth elsewhere[8,16,52], where we have shown that the gene expression data are robust to leaving a given donor out of the analysis. Briefly, we used FreeSurfer's *recon-all* to reconstruct and parcellate the cerebral cortex of each AHBA donor using the corresponding T1-weighted volume[56]. Tissue samples were assigned to the nearest parcel centroid of the left hemisphere of our parcellation in each subject's native space. For the two subjects with right hemisphere data, we first reflected the right hemisphere samples' coordinates and then performed the mapping. The median regional expression was estimated for each gene across participants ($N = 6$) and then each gene's regional values were normalized (Z-scored), resulting in a 152 (regions) × 15,043 (genes) matrix of the genome-wide expression data for the left hemisphere. The code and data underlying the AHBA alignment are available online at https://github.com/RafaelRomeroGarcia/geneExpression_Repository.

**Partial least squares regression of MS differences.** This method—applied in similar analyses integrating neuroimaging and brain gene expression data—has been described previously[8,19] (see also Supplementary Fig. 1). Here, we employ PLS regression to rank AHBA genes by their multivariate spatial alignment with cortical MS changes in each of the six different CNV conditions (+X, +Y, +21, −X, −22q11, −11p13). As detailed below, these ranked gene list for each CNV condition (Supplementary Dataset 1) provide a unifying framework to test for preferential spatial alignment between CNV-induced MS change and the spatial expression user-defined gene sets of interest (e.g. genes within vs. without the CNV region, gene sets defining different cell types etc.).

Briefly, PLS regression is a data reduction technique closely related to principal component analysis (PCA) and OLS regression. Here we use the SIMPLS algorithm[57] in R ("pls" package[58]), where the independent variable matrix ($X$) and the dependent variable ($Y$) is centered giving rise to $X_0$ and $Y_0$ respectively. The first component is then weighted by $w_1$ and $q_1$ to calculate factor scores (or PLS component scores) $T_1$ and $U_1$.

This $T_1$ is the weighted sum of the centered independent variable:

$$T_1 = X_0 w_1 + E_1, \qquad (3)$$

and $U_1$ is the weighted sum of the centered dependent variable:

$$U_1 = Y_0 q_1 + E_2, \qquad (4)$$

The weights and the factors scores are calculated to ensure the maximum covariance between $T_1$ and $U_1$, which is a departure from regular PCA where the scores and loadings are calculated to explain the maximum variance in $X_0$.

The SIMPLS algorithm provides an alternative where the matrices are not deflated by the weights when calculating the new components, and, as a result, it is easier to interpret the components based on the original centered matrices.

As the components are calculated to explain the maximum covariance between the dependent and independent variables, the first component need not explain the maximum variance in the dependent variable. However, as the number of components calculated increases, they progressively tend to explain less variance in the dependent variable. We verified that the first component ($U_1$, used for gene rank analysis) for each CNV-specific PLS explained the most relative variance.

For each CNV, we used $U_1$ to rank genes by their PLS loadings (from large positive to large negative PLS loadings, Fig. 1a). The polarity of the PLS components was fixed so that gene ranks would have the same meaning across all CNVs. Thus, for all CNV-induced MS change maps, genes with large positive PLS weights had higher than average expression in cortical regions where MS is increased in CNV carriers relative to controls (i.e., red regions in Fig. 1b), and lower than average expression in cortical regions where MS is decreased in CNV carriers relative to controls (i.e., blue regions in Fig. 1b). Conversely, genes with large negative PLS weights had higher than average expression in cortical regions where MS is reduced in CNV carriers relative to controls (i.e. blue regions in Fig. 1b), and lower than average expression in cortical regions where MS is increased in CNV carriers relative to controls (i.e. red regions in Fig. 1b). Mid-ranking genes with smaller PLS weights showed expression gradients that are weakly related to the pattern of cortical MS change.

It is important to note that $T_1$ and $U_1$ are the first PLS component weights in the common dimension of the $X$ and $Y$ variables. Thus, in our analyses comparing AHBA gene expression to cortical MS change (as in the example interpretation above), the common dimension is at the level of the nodes. However, in our analyses comparing individual patient gene expression to individual cortical MS maps, the common dimension was people rather than brain regions (see below).

**Median rank gene enrichment analysis.** The ranked gene lists provided by PLS regression of AHBA expression and MS change provided a common framework to test if the spatial expression of a given gene set was nonrandomly related to an observed spatial pattern of MS change. Specifically, we quantified this degree of spatial correspondence or a given gene set using an objective and simple measure of median gene set rank. This allowed for interpretation of gene rank enrichment both relative to the center of the rank distribution, and relative to the extremes of the list. Statistical significance of observed gene set median ranks was established by comparison with null median rank distributions from 10,000 gene rank permutations ($P_{RAND}$).

For the full chromosome CNVs, median ranks were assessed for chromosomes 1:22, X, Y, and the pseudoautosomal region (PAR, or X | Y). For plotting purposes, results with full chromosomes are presented in Fig. 1b, and results with all chromosomes and PAR genes are shown in Supplementary Dataset 2. For the subchromosomal deletions (−22q11.2/VCFS and −11p13/WAGR), we performed additional variants of our $P_{RAND}$ test ($P_{RAND\text{-}Cis}$ and $P_{RAND\text{-}Trans}$), only comparing observed median ranks to those for 10,000 from gene sets of equivalent size resampled from relevant chromosome (i.e., chromosome 22 for −22q11.2 and chromosome 11 for −11p13).

Given that CNV gene sets varied greatly in size, and the smallest gene set (+Y), was notable for being the only gene set that had an observed median ran that fell below the nominal $P_{RAND} = 0.05$ threshold, we conducted supplementary analyses to investigate the relationship between CNV gene set size and the statistical significance of observed CNV gene set median ranks relative to the $P_{RAND}$ null distribution. We decided to nest these analyses in the context of the X-chromosome, which was the CNV that contained the greatest number of linked genes in the AHBA. Across different subsamples of the X-chromosome gene set, ranging from the set size of the Y-chromosome (smallest whole-chromosome CNV) to the full size of the X-chromosome, we generated 10,000 median gene ranks from the +X PLS-ranked gene list within each subsample, as well as median gene ranks from random pulls of the entire (AHBA-overlapping) genome of comparable set size (Supplementary Fig. 2c). Since pairs of X-chromosome subsets and random subsets were arbitrarily matched, subsample $P$ values were calculated by testing the median of the X-chromosome median gene ranks against the 10,000 null median gene ranks generated by the random pulls. This was performed for each subsample size (Supplementary Fig. 2c) to evaluate a predicted $P$ value for median ranks of CNV gene sets sized similarly to the CNVs (+Y, +21) observed in our study.

Due to the fact that MS change maps integrated information from multiple individual anatomical metrics (e.g. cortical thickness, surface area, etc.), we tested if anatomical change maps for each of these individual MS features were also capable of recovering the preferential relationship between cortical expression gradients for CNV genes in health, and CNV effects on cortical anatomy in patients. To achieve this, we repeated the analytic steps detailed above for each CNV, replacing the MS change map with change maps for every individual metric used as part of our five-feature MS mapping (Supplementary Fig. 2a): gray matter volume, cortical thickness, surface area, mean curvature, and intrinsic curvature. PLS-derived gene ranks from all these analyses were assessed for statistically significant extreme ranking of CNV gene sets ($P_{RAND} < 0.05$, Supplementary Dataset 3).

**Gene ontology enrichment analyses.** Functional enrichment was assessed using rank-based gene ontology (GO) enrichment analysis. First, we subsetted the full PLS-ranked gene lists for each CNV to only contain genes that were determined as brain-expressed (see below). Then, each refined brain-only CNV gene list was inputted to GOrilla[59,60] ordered by PLS score separately in increasing and decreasing order to obtain enrichments for both tails of the gene list. Full output can be found in Supplementary Dataset 4.

**Generating cell-class gene expression maps.** We compiled data from five different single-cell studies using postmortem cortical samples in human postnatal subjects[61–65], to avoid any bias based on acquisition methodology or analysis or thresholding.

To obtain gene sets for each cell type, categorical determinations were based on each individual study, as per the respective methods and analysis choices in the original paper. All cell-type gene sets were available as part of the respective papers. For the Zhang et al.[61] and Darmanis et al.[64] papers, these data had already been reported elsewhere[66] and therefore were re-used in the present study. This approach led to the initial inclusion of 58 cell classes, many of which were overlapping based on nomenclature and/or constituent genes. The genes within each of these 58 cell-types are compiled in Supplementary Dataset 5.

We generated spatial maps of expression for each cell type gene set by calculating the median regional expression score for each gene set in the AHBA bulk microarray dataset (Fig. 2a). Then we performed hierarchical clustering of this region-by-cell-type expression matrix, using the gap statistic[22] criterion. This unsupervised analysis enabled us to determine if the cell-type gene sets from diverse studies could be grouped into biologically grounded clusters by their patterned expression across the cortical sheet. The clustering of study-specific gene sets according to known cell classes was taken to indicate that gene expression gradients in the cortical sheet are partly organized by cell-type.

The convergence of cell-type expression topography allowed us to cluster individual study cell-type gene lists into canonical cell classes. Within the context of the $N = 3$ hierarchical clustering solution from Fig. 2a, we performed post hoc assignment of each study-specific cell-type into cell classes based on the visualization of the t-Distributed Stochastic Neighborhood Embedding (tSNE) solution (Fig. 2b) on the data from Fig. 2. This solution clearly organized study-specific cell types into seven canonical classes, which were fully nested within the $N = 3$ hierarchical clustering solution from Fig. 2a. These seven classes were: astrocytes (Astro), endothelial cells (Endo), microglia (Micro), excitatory neurons (Neuro-Ex), inhibitory neurons (Neuro-In), oligodendrocytes (Oligo), and OPCs.

To derive the expression maps for each of these seven cell classes, we first collapsed across study-specific gene lists to generate a single omnibus gene list for each cell class, and then calculated a weighted average expression for each cell-class gene set in each region of our 152 AHBA parcellation (Fig. 2c). Weights for each underlying cell-type were computed by estimating the Euclidean distance of each cell-type from the centroid of their respective cell class using PCA. Two studies did not subset neurons into excitatory and inhibitory, and thus these gene sets were excluded from this cell-class assignment. Additionally, only one study included the annotation of the "Per" (pericyte) type, and thus this gene set was also excluded.

### Cortical map comparison of overall cell-class expression.
To validate the individual cell-class expression maps derived from integration of single-cell expression studies and AHBA microarray data (Fig. 2c), we computed the spatial correlation of each cell-class expression map to established maps of cortical microstructure from diverse in vivo neuroimaging and postmortem histological studies, including maps of cytoarchitecture[67] myeloarchitecture[19], and gradients of evolutionary[24], developmental[24], and interindividual (allometric) anatomical scaling[23] (Supplementary Fig. 4).

For the cytoarchitectonic maps, a 100-μm resolution volumetric histological reconstruction of a postmortem human brain from a 65-year-old male was obtained from the open-access BigBrain[67] repository on February 2, 2018 (https://bigbrain.loris.ca/main.php). Using previously defined surfaces of the layer I/II boundary, layer IV, and white matter[25], we divided the cortical mantle in supragranular (layer I/II to layer IV) and infragranular bands (layer IV to white matter). Band thickness was calculated as the Euclidean distance between the respective surfaces. To approximate cellular density, we extended upon recent work on BigBrain microstructure profiles[68] and generated microstructure profiles within supra- and infragranular bands. Intensity profiles using five equivolumetric surfaces within the predefined surfaces of the BigBrain were then averaged to produce an approximate density value. Calculations were performed at 163,842 matched vertices per hemisphere, then averaged within each cortical region in our parcellation.

The myeloarchitecture (magnetization transfer, or MT) and anatomical scaling maps used in these comparative analyses were taken from previous studies[19,23,24,67].

### Spatial permutation testing for cortical map comparisons.
To assess the specificity of the correspondence between pairs of cortical maps, we generated 10,000 rotations (i.e., spins) of the cortical parcellation[16,53]. This matching provides a mapping from the set of regions to itself, and allows any regional measure to be permuted while controlling for spatial contiguity and hemispheric symmetry.

We first obtained the spherical surface coordinates of each of our 308 regions on the fsaverage template in Freesurfer. These were then rotated about the three principal axes at three randomly generated angles. Given the separate left- and right-hemisphere cortical projections, the rotation was applied to both hemispheres. However, to preserve symmetry, the same random angles were applied to both hemispheres with the caveat that the sign of the angles was flipped for the rotation around the y and z axes.

Following each rotation, coordinates of the rotated regions were matched to coordinates of the initial regions using Euclidean distance, proceeding in descending order of average Euclidean distance between pairs of regions on the rotated and unrotated spheres (i.e., starting with the rotated region that is furthest away, on average, from the unrotated regions).

### Linking CNV-induced anatomical changes to cell-class expression maps.
Our analysis of expression gradients for previously reported single-cell expression signatures (see above) yielded an omnibus gene set for each of the seven canonical cell classes. We assessed the relationship between cortical expression of these cell classes and cortical MS change in each CNV by considering two complementary features. First, we identified cell-class gene sets that occupied significantly extreme ranks in each CNV's ranked gene list from AHBA ($P_{RAND} < 0.05$). This rank-based criterion provides a test for the degree of spatial coupling between cortical expression of each cell class and each CNV change map. Then, among the cell classes that met this rank-based criterion for a given CNV, we examined the expression of CNV genes to identify cell classes that expressed CNV genes which (i) were independently recorded as being brain-expressed from proteomic data (see below), and (ii) occupied extreme ranks (<5th, or >95th centile) alongside the cell-class gene list in the relevant CNVs ranked gene list.

### Validation against gene expression data in CNV carriers.
Dosage-sensitive (DS) genes were defined as those within the CNV region that were reported to show a statistically significant fold change in congruence with the genomic copy number change (i.e., increased in duplication carriers vs. controls or decreased in deletion carriers vs. controls).

Prior reports enabled us to define DS genes in +21 for two different tissue types: brain[30] and blood-derived LCLs[31]. Brain DS genes were defined as all chromosome 21 genes determined to show developmentally stable and statistically significant upregulation in patients vs. controls by authors of a prior study of postmortem brain tissue (see Supplementary Table 3 from ref. [30]). The LCL DS gene set was defined as all chromosome 21 genes found to be significantly upregulated in LCLs from postnatal +21 CNV carriers relative to controls (see Table 3 from ref. [31]). For each tissue, non-dosage-sensitive (nDS) chromosome 21 genes were defined as those within the AHBA dataset that did not fall within the respective tissue DS set.

For X-chromosome aneuploidies, DS X-linked genes were defined using a prior microarray study[29] of X-chromosome dosage effects on gene expression in LCLs from participants with a wide range of X-chromosome complements. X-linked LCL DS genes were defined as all X-linked genes with expression levels showing a significant positive association with X-chromosome count variation across a wide karyotype range spanning X-chromosome monosomy (i.e., −X CNV), euploidy, and X-chromosome duplication states (i.e., +X CNV). This criterion (see Supplementary Information Text S3 from ref. [29]) defined 40 DS genes for −X and +X CNVs. Non-dosage-sensitive genes for these CNV conditions were defined as all X-linked genes within the AHBA dataset that did not fall within the DS gene set.

We used median rank comparisons to test if DS and nDS genes showed patterns of cortical expression that were differentially correlated to cortical MS changes in each CNV (Fig. 3b, left). Specifically, the observed difference between median ranks of DS and nDS sets was compared to the differences of 10,000 gene rank permutations ($P_{RAND}$).

A median rank difference between two gene sets could be driven by a difference in overall rank distribution between gene sets, or by a subgroup of genes in one or both sets with extreme ranks. We used rank decile analysis to differentiate between these two scenarios. Specifically, we (i) computed the difference in the proportion of genes in the DS vs. nDS gene sets for each decile of the CNV ranked gene lists, and (ii) tested for deciles with significant differences at $P_{RAND} < 0.05$ (see Fig. 3b, right). For all four instances of DS-nDS gene set comparison (+21 brain-derived sets; +21, −X, +X LCL-derived sets), median rank differences between the DS and nDS gene set were driven by a small subset of extreme-ranked DS genes (DS$^{SS}$, Fig. 3b, c, Supplementary Dataset 6).

For all three CNVs considered in these analyses (+21, +X, −X), the median rank for these DS$^{SS}$ CNV genes was of an opposite polarity to that observed for the CNV gene set as a whole (cf. Figs. 3c, 1b). This observation implied that observed cortical MS changes in +21, +X and −X CNVs could be related to two opposing cortical gradients of CNV gene expression: those for DS$^{SS}$ genes vs. those for nDS genes. To verify this inference, we compared the cortical pattern of MS change for each of these CNVs from neuroimaging data, to the cortical pattern of differential expression for DS$^{SS}$ vs. nDS gene sets as calculated from AHBA postmortem data (Fig. 3d).

### Linking peripheral gene expression and brain anatomy.
These analyses sought to validate the relationship between CNV gene expression and cortical MS using the axis of interindividual variation. We could test the relationship between interindividual variation of gene expression and cortical MS using a subset of 55 CNV carriers in our study from whom we had gathered measures of LCL gene expression as well as sMRI brain scans. These study participants all carried an extra X-chromosome (11 XXX, 23 XXY, 11 XXYY), and originated from the National Institutes of Health Sex Chromosome Aneuploidy cohort. Details of sMRI data collection and MS map calculation for this cohort have already been described above. As part of a previously published gene expression study, we had also generated qRT-PCR (quantitative reverse transcription polymerase chain reaction) measures of gene expression in LCL tissue from these participants for 11 DS X-linked genes. These 11 genes had been selected based on a genome-wide microarray screen for X-chromosome dosage effects on LCL gene expression in sex chromosome aneuploidy conditions[29]. The methods for generation, preprocessing and analysis of these qRT-PCR data have been detailed previously[29]. Briefly, RNA was extracted by standard methods (Qiagen), and qRT-PCR was performed using the Fluidigm platform. For data processing, an assay with Ct > 23 was deemed to be not expressed. Expression data were normalized relative to the averaged expression of the two housekeeping genes ACTB and B2M, which were not differentially expressed across groups in either microarray or rtPCR data.

Before inter-relating gene expression and cortical MS across these 55 +X carriers, we first scaled gene expression and MS data across individuals within each karyotype group to remove between-karyotype group effects. This enabled us to test if, within any given +X karyotype group, greater disruption of DS gene expression was related to a cortical MS map that more strongly resembled the +X MS change map (Fig. 1b). To achieve this we used PLS regression to interrelate interindividual variation in gene expression and interindividual variation in cortical MS (see above). Partial least squares regression defined a principal component of covariance between gene expression and cortical MS across patients, and feature loadings onto this component: one for each gene, and one for each cortical region.

The cortical region loadings from this PLS component were then compared to the +X cortical MS change map in order to test of those regions which are most sensitive to X chromosome dosage are also those that vary most with interindividual variation in expression of DS X-linked genes among carriers of an extra X chromosome. This map comparison consisted of computing the spatial correlation between PLS loadings and the +X MS change map, and comparing this correlation to the distribution of 10,000 correlations given by random spatial rotations of the +X MS change map (i.e., $P_{SPIN}$).

**Defining brain-specific genes**. The genes included within the AHBA dataset cannot be assumed to be all brain-expressed. In our analytic approach of ranking genes based on the multivariate correlation (via PLS regression) between their brain expression and CNV-induced anatomical changes, high-ranking genes must show some spatial variation in their expression such that they have a non-zero expression in at least some brain regions. Not filtering the GO and single-cell analyses by brain expression would therefore risk artifactual elevation of GO terms (and cell-type enrichments) relating to brain expression.

Thus, for the GO enrichment analyses and the single-cell enrichment analyses detailed above, we first thresholded our whole-genome gene set ($N = 15,043$) to only contain genes that were determined as brain-expressed via the Human Protein Atlas (HPA; https://www.proteinatlas.org/) database of normal tissue expression. Genes whose levels of expression were not-detected (as determined by the HPA) in the cerebral cortex were excluded, yielding a list of $N = 7971$ genes with detected brain expression (Supplementary Dataset 7).

**Reporting summary**. Further information on research design is available in the Nature Research Reporting Summary linked to this article.

## Data availability
All relevant data to perform the analyses performed in this manuscript can be found here: https://github.com/jms290/PolySyn_MSNs.

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

## Acknowledgements

The authors would like to thank the patients and their families for their participation in this study. The authors also thank Francis McMahon and members of the Human Genetics Branch for helpful discussions. This work was supported by the Intramural Research Program of the National Institute of Mental Health and the *Eunice Kennedy Shriver* National Institute of Child Health (Clinical Protocol no. 89-M-006 and 08-CH-213; NIH Annual Report Numbers, 1ZIAMH002949-02, ZIA MH002794-13 and 1-ZIA-HD008898). J.S. was supported by the NIH Oxford-Cambridge Scholars' Program. F.V. was supported by the Gates Cambridge Trust. R.A.I.B. was supported by a Postdoctoral Fellowship from the British Academy. P.E.V. was supported by the Medical Research Council (MR/K020706/1) and by MQ: Transforming Mental Health (MQF17_24). She is a Fellow of the Alan Turing Institute funded under the EPSRC grant EP/N510129/1. S.E.M. was supported by a Henslow Fellowship at Lucy Cavendish College, University of Cambridge, funded by the Cambridge Philosophical Society. J.C.H. was supported by an NIH Bench-to-Bedside grant.

## Author contributions

J.S., E.T.B. and A.R. conceived of the project. J.S. performed the analyses. F.M.L., L.S.C., J.D.B., J.C.H., N.R.L., D.G.M. and A.R. were involved in data collection. A.N., S.L., R.A.I.B., P.E.V., S.E.M., R.R.-G., F.V., C.P., B.B., K.W., D.P., L.d.l.T.-U. and D.H.G. provided critical input on the methods and analyses. E.T.B. and A.R. supervised the project. J.S., E.T.B. and A.R. wrote the initial manuscript. All authors edited and contributed to the final version of the manuscript.

## Competing interests

The authors declare no competing interests.
