## [Peer Review File · Nature Communications]

Reviewers' Comments:

Reviewer #1:

Remarks to the Author:

In general, the authors diligently responded to my concerns and critiques.

However, I am still concerned about conceptual issues, especially about translating these results to major psychiatric disorders such as schizophrenia (SCZ), autism spectrum disorders (ASD) or attention-deficit disorder (i.e., ADHD) (Response 1 in the rebuttal letter).

While I understand that including severe sub-chromosomal abnormalities could help for "generalizability of findings in more than one CNV condition", I maintain the reservation concerning the generalizability to psychiatric disorders. Relative to the opinion of considering neurodevelopmental disorders as a "broad outcome category", I respectfully disagree with the authors.

In my view, this tendency would further dilute and delegitimize mental conditions already suffering from a lack of reliable criteria for diagnosis. The widespread confusion regarding SCZ, ASD and ADHD already hampered the translation of biological findings into new, more effective therapies. While I do not dismiss the importance of a known genetic entry point for understanding genetically driven disruptions of brain development, I would argue that in the case of SCZ, ASD and ADHD, this could lead also to a false trail. The reality is that CNVs and mental disorders include individuals with fundamentally different genomic landscapes: whereas the CNVs have more circumscribed and severe genetic abnormalities, at least SCZ has more subtle deficits, spread over the entire genome (see the "omnigenic" model of J. Pritchard). Needless to say that epigenetic influences will further complicate these landscapes and will contribute to the highly divergent trajectories, with distinct clinical and molecular phenotypes. I think the authors need to discuss this with more thought.

Reviewer #2:

Remarks to the Author:

My concerns have been satisfactorily addressed.

Reviewer #4:

Remarks to the Author:

The authors have successfully addressed the majority of comments the reviewer has raised.

The only comment that wasn't fully addressed is Comment 1. While I appreciate authors have put more information about details in MSN, it is still unclear how altered MS should be interpreted. For example, authors have stated, "brain regions showing relatively low expression of the causal gene set in health tended to show MS increases in patients with gene set deletion, and MS increases in gene set duplication." The pattern in Figure 1b is clear that there is an obvious link between MS change and gene ranks, but what does that change in similarity means in neuroanatomy for each disorder is unclear. Is it possible for authors to provide some information about (1) which MS feature (cortical thickness, surface area) was a major driver of MS changes for each disorder, and (2) which direction of the effect they observed (e.g. decrease in cortical thickness was observed with the increased expression level)? Or they can comment on their interpretation on what does MS change mean in terms of brain anatomy feature for each CNV disorder. This would give a clearer picture to the readers about what features in the brain anatomy was mostly affected by CNVs.

Minor comments

Page 8: 403-404: those CNV region genes that that: there are two 'that's

Reviewer 1

Comment 1

In general, the authors diligently responded to my concerns and critiques.

However, I am still concerned about conceptual issues, especially about translating these results to major psychiatric disorders such as schizophrenia (SCZ), autism spectrum disorders (ASD) or attention-deficit disorder (i.e., ADHD) (Response 1 in the rebuttal letter). While I understand that including severe sub-chromosomal abnormalities could help for “generalizability of findings in more than one CNV condition”, I maintain the reservation concerning the generalizability to psychiatric disorders. Relative to the opinion of considering neurodevelopmental disorders as a “broad outcome category”, I respectfully disagree with the authors. In my view, this tendency would further dilute and delegitimize mental conditions already suffering from a lack of reliable criteria for diagnosis. The widespread confusion regarding SCZ, ASD and ADHD already hampered the translation of biological findings into new, more effective therapies. While I do not dismiss the importance of a known genetic entry point for understanding genetically driven disruptions of brain development, I would argue that in the case of SCZ, ASD and ADHD, this could lead also to a false trail. The reality is that CNVs and mental disorders include individuals with fundamentally different genomic landscapes: whereas the CNVs have more circumscribed and severe genetic abnormalities, at least SCZ has more subtle deficits, spread over the entire genome (see the “omnigenic” model of J. Pritchard). Needless to say that epigenetic influences will further complicate these landscapes and will contribute to the highly divergent trajectories, with distinct clinical and molecular phenotypes. I think the authors need to discuss this with more thought.

Response 1

To address this reviewer’s outstanding concern, we have now altered the title, abstract and introduction of our manuscript to specifically refer to “neurogenetic” disorders, rather than the broader class of “neurodevelopmental disorders” that we had initially contextualized our work in relation to. As a result, our manuscript now reads in its framing and discussion of results as pertaining more narrowly to selective brain vulnerability in genetically-defined subtypes of neurodevelopmental disorder. Our discussion sections already refers exclusively to “neurogenetic” as opposed to “neurodevelopmental” disorders.

Key text changes made in response to Reviewer 1’s remaining concern are:

Title: *“Transcriptomic and Cellular Decoding of Regional Brain Vulnerability to Neurogenetic Disorders”*. (changed from “Neurodevelopmental Disorders”)

Abstract: *“Our work clarifies general biological principles that govern the mapping of genetic risks onto regional brain disruption in neurogenetic disorders.”*. (changed from “neurodevelopmental disorders”)

Introduction: “Clarifying factors that shape regional brain vulnerability **to genetic risks** would represent a major step forward for the translational medicine of neurodevelopmental disorders.” (clause added)

Introduction: “Recent experimental work in mice has suggested an organizing principle for regional brain vulnerability **to genetic risks** that may apply in human neurodevelopmental disorders.” (clause added)

Introduction: “Here, we conduct the first genetically-informed tests of the transcriptional vulnerability model **in humans.**” (deleted “neurodevelopmental disorders”)

Introduction: “Such grounding of regional transcriptional vulnerability in cell-type composition could provide a principled framework for nominating specific genes within specific cell-types that may account for altered anatomy in a given brain region to a given **neurogenetic disorder.**” (changed from “neurodevelopmental disorders”)

Reviewer 4

Comment 1

The authors have successfully addressed the majority of comments the reviewer has raised.

The only comment that wasn't fully addressed is Comment 1. While I appreciate authors have put more information about details in MSN, it is still unclear how altered MS should be interpreted. For example, authors have stated, “brain regions showing relatively low expression of the causal gene set in health tended to show MS increases in patients with gene set deletion, and MS increases in gene set duplication.” The pattern in Figure 1b is clear that there is an obvious link between MS change and gene ranks, but what does that change in similarity means in neuroanatomy for each disorder is unclear. Is it possible for authors to provide some information about (1) which MS feature (cortical thickness, surface area) was a major driver of MS changes for each disorder, and (2) which direction of the effect they observed (e.g. decrease in cortical thickness was observed with the increased expression level)? Or they can comment on their interpretation on what does MS change mean in terms of brain anatomy feature for each CNV disorder. This would give a clearer picture to the readers about what features in the brain anatomy was mostly affected by CNVs.

Response 2

We have now added new analyses and results (new **Fig. S2b** panel) that directly address the reviewer's request to more directly show “which MS feature (cortical thickness, surface area) was a major driver of MS changes for each disorder”. The manuscript text and figure panel relating to these new analyses are reproduced below.

Results. Mapping altered cortical anatomy in 6 different CNV conditions : “In order to test for differential contribution of single cortical features to observed maps of regional MS change in each CNV, we (i) recomputed CNV-specific MS change maps with exclusion of each individual cortical feature prior to MSN construction and then (ii) determined which of these single feature exclusions most change the topography of observed MS change for each CNV. This leave-one-out procedure showed that mean curvature (+X, +Y) and gray matter volume (-X,+21,-22q11,-11p13) were the features that most contributed to the topography of observed MS changes (**Fig. S2b**).”

New FigS2 (old S3)

a.**b.****c.**
Fig. S3. Gene set size and constituent features of neuroanatomical effects. **a)** Regional T-statistics (CNV patients vs. controls) computed using individual constituent features of the morphometric similarity networks. The CNV gene set ranks linked to these alternative anatomical change maps are provided in Table S3. Black outlines denote anatomical change maps which successfully recover the preferential spatial coupling between anatomical change and expression of CNV region genes in the human cortex ($P_{RAND} < 0.05$). **b)** Heatmap showing the correlation between MS change maps computed with all morphological features (“MSN”) and those with one feature left out. Hashes denote minimum correlations with the original MS change map. **c)** Raincloud plot demonstrating consistency of effect-size between size-varying subsamples of X-chromosome gene ranks for the +X CNV (blue), and random gene subsamples of similar size (red). Boxplots show the median and interquartile ranges. Median ranks are highly consistent, but variation in gene rank increases with reducing gene set size. Black asterisks denote significance of median differences, with a ~100 genes being the smallest gene set size necessary to consistently reach significance at $P_{RAND} = 0.05$. Gray annotations denote the +Y and +21 CNVs, showing that they fall within the expected trend for significance based on gene sets of similar size within the X-chromosome.

Comment 2

Page 8: 403-404: those CNV region genes that that: there are two ‘that’s

Response 2

This has been corrected, thank you.

Reviewers' Comments:

Reviewer #1:

Remarks to the Author:

The authors have satisfactorily responded to my suggestions.

Reviewer #4:

Remarks to the Author:

Authors have addressed all the issues. I believe the paper is now ready for publication.